# Contralateral C7 Nerve Transfer for Stroke Recovery: New Frontier for Peripheral Nerve Surgery

**DOI:** 10.3390/jcm10153344

**Published:** 2021-07-29

**Authors:** Ali M. Alawieh, Nicholas Au Yong, Nicholas M. Boulis

**Affiliations:** Department of Neurosurgery, Emory School of Medicine, Emory University, Atlanta, GA 30322, USA; nicholas.au.yong@emory.edu

**Keywords:** stroke, C7 transfer, peripheral nerve, neurorehabilitation

## Abstract

Ischemic stroke remains a major cause of disability in the United States and worldwide. Following the large-scale implementation of stroke thrombectomy and the optimization of treatment protocols for acute stroke, the reduction in stroke-associated mortality has resulted in an increased proportion of stroke survivors, many of whom have moderate to severe disability. To date, the treatment of subacute and chronic stroke has remained a challenge. Several approaches, involving pharmacological interventions to promote neuroplasticity, brain stimulation strategies and rehabilitative interventions, are currently being explored at different stages of the translational spectrum, yet level 1 evidence is still limited. In a recent landmark study, surgical intervention using contralateral C7 nerve transfer, an approach used to treat brachial plexus injury, was implemented in patients with chronic stroke, demonstrating an added benefit to standard rehabilitation strategies, leading to improved motor performance and reduced spasticity. The procedure involved the transfer of the C7 nerve root and middle trunk from the uninjured extremity to the injured extremity using a short conduit that allows for faster regeneration and innervation of the injured upper extremity via the ipsilateral (contralesional) hemisphere. In this work, we review the rationale for using contralateral C7 nerve transfer in stroke, describe the surgical intervention with associated variations and limitations, and discuss the current evidence for the efficacy of this technique in ischemic stroke research.

## 1. Introduction

Although mortality from stroke has significantly decreased over the past decade secondary to improved preventative efforts and the large-scale implementation of thrombectomy techniques, stroke continues to be the fifth major cause of death in US and a leading cause of disability among adults [1,2,3]. Despite significant improvement in stroke prevention and the acute management of large vessel occlusion over the past 2 decades, clinical management and research involving the subacute and chronic phases of recovery is still lagging behind. Given the increased number of stroke survivors, there is a growing need for novel pharmacological, surgical and rehabilitative strategies that can expand and leverage the window of neuronal plasticity to enhance chronic functional recovery. Currently, there are no therapeutic agents yet that have been shown to confer neuroprotection beyond the acute phase, there are no standard neurorehabilitation protocols for optimizing recovery, and there are no surgical interventions that are routinely used for stroke recovery. 

## 2. Leveraging the Contralesional Hemisphere in Motor Recovery after Stroke

Despite the myriad of deficits that result from stroke, including motor, sensory, speech and cognitive deficits; motor deficits remain the most recognized deficits with the highest impact on the quality of life of patients and caregivers [3,4]. Anatomically, three major pathways have been adopted to facilitate post-stroke motor recovery, namely, promoting contralateral (ipsilesional) cortical/subcortical recovery, promoting compensatory pathways, recruiting ipsilateral (contralesional) circuitry to take over functions of the injured hemisphere, or targeting subcortical and cerebellar postural circuitry to improve gait and postural stability. 

Neuromodulatory strategies targeting the injured hemisphere range from pharmacological interventions to inhibit the expansion of injury and inflammation, to invasive and non-invasive neurostimulation of the injured hemisphere to induce neuronal re-organization and reinforce salvageable circuitry, and motor rehabilitation and skills training to develop compensatory recovery mechanisms [3,5,6]. However, the window of neuronal plasticity and salvageable neuronal substrates in the injured hemisphere is still limited and is often not an optimal target in chronic stroke patients who have missed the window for post-stroke neuroplasticity and whose damage to the injured hemisphere is now stable. Therefore, there has been emerging interest in leveraging the neuronal substrate within the contralesional (unaffected) hemisphere, to promote ipsilateral motor functional recovery [7]. Neuromodulation of the ipsilateral hemisphere after stroke has not yet shown promising clinical results, and outcomes have been variable across studies [8,9]. Although it has been proposed that an ipsilateral corticospinal tract pathway could mediate these ipsilateral connections, critical review of data from primates and humans fails to demonstrate a clinically relevant ipsilateral corticospinal tract pathway (reviewed in detail in [7]). In contrast, evidence suggests that the predominant ipsilateral motor input is via modulation of the contralateral hemisphere via inter-hemispheric inhibition, callosal projections or via modulation of motor activity by the ipsilateral somatosensory and association cortices [9]. Direct connection from the motor cortex to ipsilateral motor neurons remains controversial and is not supported by current literature. Finally, more recent approaches have investigated modulation of the motor circuitry at the level of subcortical and cerebellar loci to improve overall functional status. Work by Machado et al. reported the possibility of targeting the dentate nucleus of the cerebellum to improve post-stroke recovery and responses to rehabilitation therapy given the robust efferent connections to the cerebral cortex [10]. Despite promising preclinical work and early human work using different neurostimulation strategies, high-quality data supporting the efficacy of these interventions in humans is still lacking, and the design of randomized controlled trials is contingent upon the optimization of stimulation targets, timing of interventions and stimulation parameters. 

Given that physiological or latent ipsilateral motor connections may be limited in humans, a novel and alternative approach is to leverage the ipsilateral uninjured hemisphere at the peripheral nervous system level, using peripheral nerve transfer. In a relevant clinical application to stroke, patients with brachial plexus injury have been treated surgically with nerve transfers, specifically C7 nerve root transfer, for several decades to help improve function and reduce spasticity in the affected limb [11,12,13,14]. The C7 nerve root is an ideal donor in these cases given that, among the C5-T1 nerve roots that form the brachial plexus, the C7 root comprises 20% of all fibers and its motor function overlaps with all four remaining roots [11,12,13,14,15]. Therefore, following a C7 nerve transection or transfer, the donor arm will show transient weakness and sensory losses that tend to recover spontaneously. Therefore, in a recent landmark study, Zheng et al. proposed the use of contralateral C7 transfer for the treatment of motor deficits and spasticity after chronic stroke in adults [16], as this approach was well tolerated in brachial plexus injury patients undergoing an analogous procedure. This new direction for neurosurgical interventions in stroke rehabilitation from the perspective of peripheral nerve surgery continues to garner interest in the clinical community and among patients. Consequently, it is important for clinicians to understand the physiology, rationale, benefits and limitations of this approach. In this work, we will review the available data on the safety and efficacy of C7 nerve transfer in stroke, the technique, and the associated neurophysiological changes following transfer. 

## 3. Contralateral C7 Nerve Root Transfer Technique

The procedure for C7 nerve root transfer has been extensively described in the peripheral nerve surgery literature for the treatment of brachial plexus injury [11,12,13,14,15,16]. In the absence of data from randomized clinical trials on contralateral C7 nerve root transfer, evidence supporting improved hand function after transfer is limited to small sample size pilot studies and case series with a wide range of variability in surgical approaches. 

The choice of the C7 nerve root as a donor root is based on prior work from brachial plexus injury demonstrating near complete recovery of sensory and motor deficits of the donor arm after transfer, due to overlapping innervation with the remaining brachial plexus root. The C7 root fibers are transected at the level of the divisions with meticulous attention to avoid injury to the nearby lateral and posterior cords. On the recipient side, the C7 nerve root contributes to the innervation of several muscles including the latissimus dorsi and pectoralis major (shoulder), triceps (forearm), extensor carpi radialis (wrist) and extensor digitorum (hand) [17]. 

In a recent systematic review of 39 contralateral C7 transfer studies in traumatic brachial plexus injury [18], a total of 754 cases were reported. The median nerve was the primary recipient in the majority of cases (60%), of which around 50% of patients were able to perform active movement, at least against gravity (Medical Research Council (MRC) score of 3 or more), in wrist flexion and finger flexion [18] and a similar proportion of patients reported sensory improvement along the median nerve distribution. Alternative recipient nerves included the musculocutaneous nerve (20% of cases), for which around two thirds of patients achieved a motor strength of at least antigravity in elbow flexion (MRC 3–4). The radial nerve or triceps branches were the recipients in 10% of cases, resulting in around a third of patients regaining strength in elbow or wrist extension at the MRC3–4 level. Alternative transfer recipients included upper or lower trunks, lateral or posterior cords, a thoracodorsal nerve, an axillary nerve, a suprascapular nerve and an ulnar nerve [18]. Even when the recipient was the median nerve, the most common recipient, there were still different versions of the procedure, with associated variability in outcomes that was mainly dependent on the distance required for the nerve to regenerate, which traditionally required long-distance ulnar nerve grafts [19,20]. In early studies, the injured side ulnar nerve was dissected down to the wrist, freed and passed across the chest to be coapted to the distal C7/middle trunk. The Tinel sign was then used to track the progression of axonal regeneration, followed by a second operation to transect the ulnar nerve and transfer to the recipient. Different versions of the technique have been optimized, requiring different types of synthetic or autologous nerve grafts of variable sizes, with ongoing efforts to try to minimize the distance required for regeneration. An additional limitation that would also limit the success of regeneration is the presence of two neurorrhaphy sites in the majority of these procedures [21]. Therefore, to optimize the success of this procedure in the context of stroke recovery, Zheng et al. used a modified version of this procedure that involves direct coaptation of the C7 donor nerve root and the recipient without the use of a nerve graft [16]. This reduces the distance required for regeneration and limits the number of coaptation bridges to one [16]. The approach used by Zheng et al. is illustrated in Figure 1 and further described below. 

The C7 nerve root from the unaffected arm (blue) is dissected and transected proximally to where it joins the remaining fibers of the brachial plexus. The C7 nerve root on the affected side (orange) is then transected at the exit from the neural foramina. The donor C7 root is then tunneled via one of multiple approaches (with or without a sural nerve graft), to the contralateral side and coapted to the recipient distal end of the C7 root on the affected side.

To perform the transfer, a supraclavicular incision is usually used, followed by dissection down to the anterior and middle scalene muscles. Following brachial plexus exposure on the recipient end, the C7 nerve root is then dissected on the affected side. The C7 root exists through the intervertebral foramen and forms the isolated middle trunk before splitting at the level of the divisions to contribute to both posterior and lateral cords. The C7 nerve root is identified at its exit from the neural foramina and confirmed via electromyogram (EMG) recordings. At the injured site, the C7 root is dissected proximally to the level of the intervertebral foramen and then transected to allow for a maximal length for transfer. Then, the brachial plexus is exposed on the donor side. The C7 root is then dissected distally until the level of the brachial plexus divisions. The nerve is then transected around 4–6 cm from the neural foramen, at the confluence of the divisions with the adjacent nerves, with careful attention to avoid injury to the posterior and lateral cords. A tunnel is then dissected anterior to the vertebral body to tunnel the donor nerve to the contralateral side. In the most recent stroke trial, the tunnel was developed using the prespinal route (between the esophagus and the vertebral body, medial to the longus colli muscle), which allows for the shortest distance for transfer [16]. Alternatively, several routes have been described to tunnel the donor C7 root including the retropharyngeal space, posterior to the anterior scalene, subcutaneously or through a retrosternocleidomastoid approach [17,22,23]. The closer the tunnel is to the vertebral body, the shorter the required graft segment is and the faster the neurotization and regeneration will occur. The approach used in the Zheng trial did not require a sural nerve graft due to the use of a prespinal route. However, close proximity to the vertebral body via the prespinal route carries the additional risk of possible injury to the vertebral artery or development of esophageal fistula.

In the event of an alternative route being chosen and to allow for a tension free suture, sural nerve grafts may be used. Around 3–4 strands of sural nerve graft are commonly used to cover the distance needed to coapt the donor to recipient nerve roots [16,17]. The donor-sural nerve coaptation is completed under an operating microscope and using 9-0 nylon suture to approximate the cut ends of the grafts and the donor divisions, ideally using two grafts per division. The graft is passed through the subcutaneous tunnel already dissected. Finally, the opposite end of the donor C7 nerve root or grafts are coapted to the recipient nerve root. This step completes the procedure, and patients are left with an immobilizing splint of the recipient arm for around 1–2 months. The overall procedure is illustrated in Figure 1. Based on cadaveric studies, the distance between the C7 nerve root and the muscle entry zone is between 10 cm (pectoralis major) and 24 cm (triceps) [23]. In their work on stroke patients, Zheng et al. used the single C7 root as recipient; however, using multiple recipient roots as opposed to a single C7 root could be an alternative approach to improve the success rate of the procedure. 

Importantly, unlike the brachial plexus injury population, the anatomy of the recipient brachial plexus is pristine in stroke patients. This fact means that proximal coaptation is technically less challenging. In addition, the recipient axons have often been traumatized distally, making regeneration less likely. The distal plexus is untraumatized and contains healthy adult axons, making them better candidates to support regeneration histologically. There is a third reason that outcomes in stroke patients may be better than in plexus trauma. It has long been understood that the capacity of recipient nerves to support axonal regeneration is inversely proportional to the time from injury. The fact that the recipient undergoes axotomy at the time of repair makes successful regeneration more likely. 

## 4. Contralateral C7 Nerve Transfer in Human Stroke

In the context of chronic stroke, motor recovery and routine activities of daily living (ADLs) are significantly impaired by both paralysis and spasticity in the affected arm, which are secondary to loss of upper motor neuron input. Although C7 nerve transfer has been robustly used and reported in brachial plexus injury, only one randomized controlled trial for contralateral C7 transfer has been published to date in the context of stroke. The trial was reported by Zheng et al. in the New England Journal of Medicine in 2018. In this trial, a total of 36 patients who are in the chronic phase of unilateral cerebral injury were randomized into a C7 transfer group versus control (18 patients each). Both groups were subjected to an extensive rehabilitation program over 1 year that involved occupational therapy, physical therapy, skills training and the use of orthoses as needed. However, the main limitation of the study was the lack of a detailed description of the rehabilitation program used in the trial, or whether the rehabilitation protocol was standardized or variable across patients. Both treatment groups included patients who were 15 years away from their initial injury on average, and they were not limited to stroke but included traumatic brain injury (a third of each group). Both groups had comparable Fugl-Meyer and Ashworth scores at baseline. The former assesses function, and the latter scores spasticity. Both patient groups were followed over a year for their functional recovery and electrophysiological outcomes [16]. 

Evidence of improvement in the surgical group was first noted in the spasticity scores (Ashworth scale) that started within the first few days of the procedure. A total of 23–50 degrees of improvement in spasticity were noted across the different joints in the affected arm by 12 months in the C7 transfer group compared to 0–1 degrees in the control arm. The most prominent improvement in spasticity (~50 degrees) was noted at the wrist in the surgical group. Given the fact that improvement in spasticity scores happened before the expected time for the nerve regeneration (~6–8 months) of the C7 fibers to the injured arm, the improvement in spasticity is likely related to the process of C7 root transection in the affected arm. 

When assessing function using the Fugl-Meyer score, a sensitive measure of motor function, there was an average of a 17-point improvement in scores by 12 months in the surgical group compared to 2.6 in controls [16] (the maximal score on the Fugl-Meyer test is 66). In patients with C7 transfer, significant improvement in functional scores correlated with the timing of expected regeneration, as the investigators demonstrated a significant improvement starting at 10 months after surgery. These improvements in motor scores were not insignificant, as patients also demonstrated improved performance in their ADLs in the surgical group, including the fact that most surgical patients were able to perform more than three ADL tasks with their affected arm by 1 year. This improvement is a combination of reduced spasticity, improved response to rehabilitation and improved power in the previously paralyzed arm. At the electrophysiological level, transcranial magnetic stimulation to the ipsilateral hemisphere demonstrated responses in the ipsilateral arm by the 10th month after surgery, and functional Magnetic Resonance Imaging (fMRI) recordings demonstrated ipsilateral activation of the motor cortex after the active movement of the affected arm. Both of these pieces of information support the creation of neuromuscular junctions connected to the contralateral C7 motor neurons. All the outcome measures were focused on motor recovery, and sensory changes in the recipient arm were not investigated. Of note, there were no major permanent adverse effects in the surgical group, with all the sensory and motor impairments noted in the donor arm nearly resolved by the end of the study. Patients in the surgery arm still complained of a foreign sensation during swallowing.

Despite the significance of this study, there remain some major limitations that need to be addressed in future studies. The cohort of patients was heterogenous and included patients with cerebral palsy and encephalitis, and at the same time, the sample size was relatively small (18/group) for such a heterogenous population. Additionally, the average age of patients was relatively young (27 years), whereas the average age for stroke patients is within the range of 50–70 years of age. A major consideration in this study was the fact that spasticity scores immediately improved in almost all the patients after surgery, an effect likely related to C7 nerve transection. An improvement in spasticity alone is likely to improve function in a paralyzed arm and could explain, at least in part, the improvement in the Fugl-Meyer scores. Therefore, future studies should include a cohort of C7 neurotomy alone to assess whether the full extent of recovery could be achieved with this component of the procedure. In that case, the surgical intervention would incur significantly lower risks. 

Following their trial, the same group then investigated the effects of the C7 nerve transfer on lower extremity spasticity and demonstrated evidence of decreased spasticity in the affected lower extremity. However, the outcome was limited to the penetration angle of the gastrocnemius muscle and the plantar load, two indirect measures of subtle changes in lower extremity spasticity in the absence of clinical improvement. Although significant from a statistical standpoint, the effect on lower extremity function remains of indeterminate clinical significance [24]. Additionally, the underlying pathophysiological mechanism of this effect remains unclear, but the authors propose that proprioceptive signaling via a previously described cervicolumbar reflex could be implicated in this process as described by Delwaide et al. [25].

To date, these studies have not yet been replicated in the Western world, but several case reports from China have reported similar findings at single case level [26,27] and described alternative routes for the development of safe short tunnels for transfer, including a posterior spinal approach by Guan et al. using a posterior approach involving a C7 laminectomy [26]. 

Finally, the work of the authors again emphasizes the significance of rehabilitation therapy after stroke, given that all the participants were subjected to an intense rehabilitation program; however, the control arm demonstrates how the current rehabilitation strategies have a limit at which the effects become saturated and minimal improvement can be achieved in the chronic phase of injury. When combined with restorative strategies such as C7 transfer, the rehabilitation clock may be re-set and a better response to rehabilitation is expected. However, the authors did not investigate the effect of C7 transfer with and without rehabilitation to determine this effect, but such an investigation is likely to not be feasible from an ethical standpoint.

## 5. Conclusions and Perspective

At a time when rehabilitation interventions and neurostimulation strategies for stroke recovery have stagnated, a novel surgical approach of contralateral C7 nerve transfer for treatment of upper extremity spasticity provides a new therapeutic approach in a time window previously thought to have limited potential for intervention. Furthermore, contralateral C7 nerve transfer can synergize with current standards of care. Following the pioneer study by Zheng et al. in 2018, there are still many unanswered questions regarding the physiological underpinnings, clinical efficacy and morbidity for this procedure. There remains a sincere need for the replication of this study using large multicenter cohorts in countries with different clinical resources and practice preferences. Another major limitation is that this procedure requires high technical neurosurgical skills and may be associated with significant morbidity in the event of complications. Of critical importance is the fact that, without a concerted rehabilitation program following surgery, the results are likely to be disappointing.

## Figures and Tables

**Figure 1 jcm-10-03344-f001:**
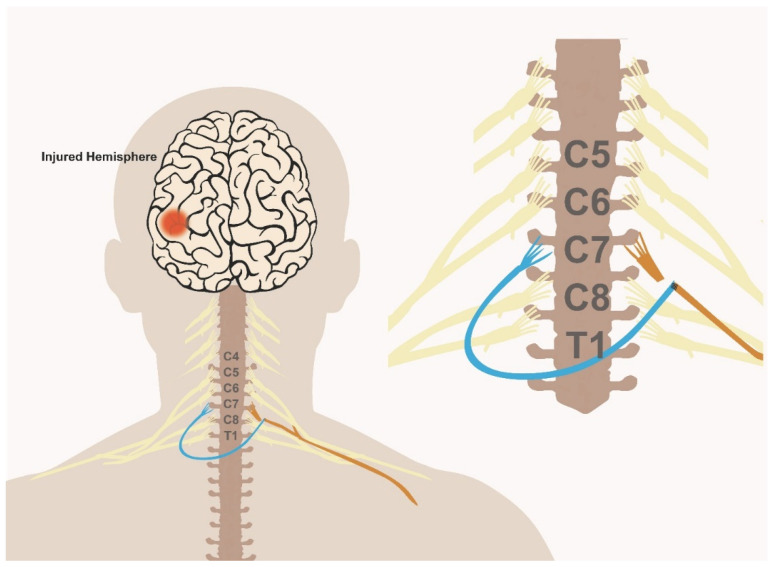
Schematic illustration of the C7 nerve transfer procedure.

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
