# Peer review of "Contralateral C7 Nerve Transfer for Stroke Recovery: New Frontier for Peripheral Nerve Surgery"

_jcm, 2021, doi:10.3390/jcm10153344_

Round 1

Reviewer 1 Report

The article reviews the current state of the art of the contralateral C7 nerve transfer for the treatment of post-stroke spasticity and weakness. In brief, the C7/middle trunk is harvested along the unaffected brachial plexus and transferred to any of a variety of targets within the brachial plexus of the paralyzed arm. Spasticity is relieved immediately with the nerve transection, while motor recovery follows several months later as the donor axons grow into their target muscles.

Compared to brachial plexus injury patients undergoing the identical procedure, stroke patients theoretically should have a better outcome given that the target nerves are un-injured and the target muscles are not denervated until the procedure is performed. The procedure, however, is complicated, and requires bilateral brachial plexus dissections and usually a neck dissection. There are a number of complications that can occur, although the available studies indicate a reasonable safety profile. One key point is that intensive rehabilitation is required for patients to optimize their outcomes after the nerve transfer surgery. Therefore, the high level of surgical and rehabilitation expertise needed to perform this procedure will limit its use to select medical centers.

This therapy may be considered in the comprehensive, multidisciplinary treatment of patients suffering from stroke with residual upper extremity paralysis.

On Page 2, Line 78-9, “In a relevant clinical application to stroke, patients with brachial plexus injury have been treated surgically with nerve transfers, specifically C7 nerve root transfer, for several decades to help improve function and reduce spasticity in the affected limb.” The term “paralysis” is probably more appropriate than “spasticity” here.

There are some minor additional typos that can be corrected at an editorial level.

Author Response

We thank the reviewer for the detailed review of our manuscript. We have performed a full editorial check on the most recent version of the manuscript. However, regarding Page 2 Line 78-81, the use of term "spasticity" rather than "paralysis" was intentional. The statement mentions that in brachial plexus injury, contralateral C7 transfer improves overall function (which is via restoration of some motor activity) but also reduced spasticity in the affected arm that often limits rehabilitation therapy and recovery. 

Reviewer 2 Report

well written overview, however hardly own data

important "hot topic" in the peripheral nerve field

please change "anastomosis" to "coaptation"

Author Response

We thank the reviewer for the positive feedback regarding our manuscript. We have changed the word anastomosis to coaptation throughout the manuscript as suggested.